# Inhibition of TPI1 Sensitizes Cisplatin-Resistant Oral Cancer to Ferroptosis

**DOI:** 10.3390/biomedicines13051225

**Published:** 2025-05-19

**Authors:** Dandan Wang, Huimin Zheng, Yumin Chen, Jialin Hao, Yuan Zhou, Nan Li

**Affiliations:** 1Department of Pediatric Dentistry, Peking University School and Hospital of Stomatology, Beijing 100081, China; wangdandankq@pkuss.bjmu.edu.cn; 2Department of Geriatric Dentistry, Beijing Laboratory of Biomedical Materials, Peking University School and Hospital of Stomatology, Beijing 100081, Chinawren1996@163.com (Y.C.); 3Department of Prosthodontics, Stomatological Hospital and Dental School of Tongji University, Shanghai Engineering Research Center of Tooth Restoration and Regeneration, Shanghai 200072, China; 4Shenzhen Stomatological Hospital, Southern Medical University, 1092 Jianshe Road, Luohu District, Shenzhen 518001, China; 5Department of Stomatology, Shenzhen People’s Hospital, Shenzhen 518020, China; 6Second Clinical Medical School of Jinan University, First Affiliated Hospital of Southern University of Science and Technology, Shenzhen 518020, China

**Keywords:** ferroptosis, TPI1, cisplatin-resistant, oral cancer

## Abstract

**Background:** Iron metabolism has emerged as a critical factor in cancer biology, with elevated intracellular iron levels contributing to increased oxidative stress and tumorigenesis. However, the molecular determinants governing ferroptosis sensitivity remain incompletely understood. Triosephosphate isomerase 1 (TPI1), a key glycolytic enzyme, has been implicated in cancer progression, but its role in ferroptosis regulation, particularly in the context of chemoresistance, is largely unexplored. In this study, we investigated the impact of TPI1 silencing on ferroptosis in cisplatin-resistant oral squamous cell carcinoma (OSCC), aiming to elucidate its mechanistic role and therapeutic potential. **Methods:** We conducted in vitro and in vivo analyses to evaluate the functional consequences of TPI1 knockdown in cisplatin-resistant OSCC cell lines and tumor xenograft models. The effects of TPI1 silencing and/or cisplatin treatment were assessed with respect to cell proliferation, migration, and invasion, along with ferroptosis-associated markers, including lipid ROS, free iron levels, lipid peroxidation, and the expression of key ferroptosis-related genes. Additionally, we analyzed the clinical relevance of TPI1 expression in human OSCC tissue samples, examining its association with clinicopathological features and patient prognosis. **Results:** TPI1 was found to be significantly upregulated in both OSCC tissues and cell lines, and high TPI1 expression correlated with poor clinical outcomes. Multivariate analysis identified TPI1 as an independent prognostic factor for tumor progression. Functionally, TPI1 knockdown suppressed OSCC cell proliferation, migration, and invasion, while its overexpression enhanced these oncogenic behaviors. Mechanistically, silencing TPI1 led to increased intracellular ROS accumulation, elevated free iron, and enhanced lipid peroxidation, collectively promoting ferroptotic cell death in cisplatin-resistant OSCC cells. In vivo, TPI1 depletion resulted in marked tumor growth inhibition and synergized with cisplatin to further suppress tumor burden in xenograft models. Moreover, TPI1 silencing disrupted the epithelial–mesenchymal transition (EMT), a key driver of cancer metastasis and drug resistance. **Conclusions:** Our study reveals a previously unrecognized role of TPI1 in protecting oral cancer cells from ferroptosis, especially in the setting of cisplatin resistance. These findings suggest that TPI1 not only contributes to tumor aggressiveness but also mediates resistance to ferroptosis. Targeting TPI1 may therefore represent a promising therapeutic strategy to overcome chemoresistance and enhance ferroptosis-based therapies in oral cancer.

## 1. Introduction

Oral squamous cell carcinoma (OSCC), a major subtype of head and neck cancer, remains a significant global health burden characterized by high morbidity and mortality. Its development is multifactorial, with well-established risk factors, including tobacco use, excessive alcohol consumption, betel nut chewing, and persistent infection with high-risk human papillomavirus (HPV) strains [1,2,3]. Clinically, OSCC often presents as non-healing ulcers, proliferative masses, cystic lesions, or, less commonly, melanomas [2]. Despite advances in diagnostic and therapeutic strategies, surgical resection remains the mainstay of treatment, and the five-year survival rate has shown only modest improvement over the past decades [3,4,5].

To improve therapeutic outcomes, chemotherapeutic agents such as 5-fluorouracil, hydroxyurea, platinum compounds (including cisplatin), and the EGFR inhibitor cetuximab are frequently utilized in combination with surgery and/or radiotherapy [6,7,8]. However, the long-term efficacy of chemotherapy is significantly hindered by the emergence of drug resistance, which contributes to poor prognosis and high recurrence rates.

Recently, ferroptosis, a distinct form of regulated cell death driven by iron accumulation and lipid peroxidation, has gained attention as a promising strategy for overcoming chemoresistance in cancer [9,10,11,12]. Several ferroptosis inducers—including erastin, sulfasalazine, and glutamate—have been shown to synergize with radiotherapy and chemotherapy, enhancing their anticancer effects while minimizing systemic toxicity [13,14,15,16]. The inhibition of key ferroptosis regulators, such as glutathione peroxidase 4 (GPX4), promotes selective cancer cell death in vitro and prevents tumor recurrence in vivo [14,17]. Notably, ferroptosis has been demonstrated to sensitize tumors to chemotherapy in various malignancies, including small-cell lung cancer, hepatocellular carcinoma, renal cell carcinoma, and pancreatic cancer [15,18]. For example, sulfasalazine enhances cisplatin efficacy by targeting the xCT transporter in colorectal cancer [17] and mitochondrial pyruvate carrier 1 modulates ferroptosis susceptibility through epithelial–mesenchymal transition (EMT) pathways in drug-tolerant head and neck cancers [11,19]. However, the role of ferroptosis in modulating cisplatin sensitivity in oral cancer remains largely unexplored.

Triosephosphate isomerase 1 (TPI1) is a key glycolytic enzyme involved in the reversible interconversion of dihydroxyacetone phosphate and glyceraldehyde-3-phosphate [20,21]. Beyond its classical role in energy metabolism, recent evidence has implicated TPI1 in various oncogenic processes, including proliferation, metastasis, and metabolic reprogramming [21,22]. TPI1 is overexpressed in multiple cancers, such as intrahepatic cholangiocarcinoma, gastric, lung, and prostate cancers, and its elevated expression has been correlated with poor clinical outcomes [21,23,24,25]. Interestingly, TPI1 functions as a serum biomarker in several malignancies, including breast cancer, lung squamous cell carcinoma, and hepatocellular carcinoma (HCC), although conflicting evidence has also suggested a tumor-suppressive role in HCC [22,26,27]. These divergent findings underscore the complexity of TPI1’s role in cancer biology and the necessity to investigate its function in specific tumor contexts.

In this study, we identify TPI1 as a novel regulator of ferroptosis in cisplatin-resistant OSCC. We demonstrate that TPI1 is overexpressed in oral cancer tissues and cell lines and that high TPI1 expression correlates with worse patient outcomes. Mechanistically, silencing TPI1 induces ferroptosis through increased lipid peroxidation and ROS accumulation, sensitizing cisplatin-resistant OSCC cells to chemotherapy. These findings not only establish TPI1 as a prognostic biomarker but also suggest its therapeutic potential in overcoming drug resistance via ferroptosis induction.

## 2. Methods and Materials

### 2.1. Bioinformatics Analysis

This study utilized the TCGA, UALCAN, and METABRIC databases, along with the Kaplan–Meier plotter and survival R packages for clinical outcome analysis. Additionally, Linkedomics, cBioPortal, and GEPIA were employed, using R version 4.0.5.

### 2.2. Tissue Specimens and Patients

Six pairs of fresh oral tissues (oral cancer and adjacent normal samples) were randomly collected for Western blot analysis. Additionally, 28 normal and 107 oral cancer samples were obtained from patients who underwent surgery in 2007 for tissue microarray studies, with follow-up until July 2021. Overall survival (OS) was measured from surgery to death. Clinical samples were collected at our hospital and confirmed by histological and pathological exams. The Ethics Committee of Peking University School and Hospital of Stomatology approved this study, and all patients signed written informed consent. Approval number: PKUSSIRB-202498056

### 2.3. Immunohistochemistry (IHC)

Tumor tissue sections (5 μm) were deparaffinized, rehydrated, and subjected to antigen retrieval with a citrate buffer. Endogenous peroxidase was blocked with 3% H2O2. Oral cancer gene chips were dewaxed, hydrated, and incubated overnight at 4 °C with a primary antibody against PD-TPI1 (1:200, Proteintech, 10713-1-AP, Beijing, China). The following day, sections were incubated with goat anti-rabbit/mouse IgG at 37 °C for 20 min, washed three times with PBS, treated with DAB substrate (Zsbio, ZLI-9018, Beijing, China), and stained with hematoxylin (Solarbio, G1080, Beijing, China). Three pathologists performed IHC scoring to assess positive cell proportions and staining intensity. Proportion scores ranged from 0 (<10%) to 4 (>75%), while staining intensity scores ranged from 0 (no staining) to 3 (dark brown). The H score was calculated as follows: (percentage of cells with score 1 × 1) + (percentage of cells with score 2 × 2) + (percentage of cells with score 3 × 3), with a maximum of 300.

### 2.4. Cell Culture

Human oral keratinocytes (HOKs) and oral cancer cell lines (CAL27, SCC15, and SCC9) were sourced from the American Type Culture Collection. They were cultured in DMEM with 10% FBS, 1% streptomycin, and 1% penicillin at 37 °C in 5% CO_2_ and 95% humidity, then randomly assigned to NT (non-cisplatin treatment) and cisplatin groups. Cells of the cisplatin group were treated with cisplatin (1 mM) for 8 h. The cells of the NT group were treated with a normal medium.

### 2.5. Chemical Inhibitors and Treatments

To investigate the specific mode of cell death induced by TPI1 knockdown, we employed a panel of small-molecule inhibitors targeting distinct regulated cell death pathways. Ferrostatin-1 (Fer-1, Sigma-Aldrich, F6696, Beijing, China) was used to inhibit ferroptosis at a final concentration of 2 μM. Z-VAD-FMK (Selleck, S7023, Beijing, China), a pan-caspase inhibitor targeting apoptosis, was used at 20 μM. Necrostatin-1 (Nec-1, Sigma-Aldrich, N9037, Beijing, China), a RIPK1 inhibitor for necroptosis inhibition, was used at 20 μM. 3-Methyladenine (3-MA, Selleck, S2767, Beijing, China), an autophagy inhibitor, was used at a concentration of 5 mM.

Cisplatin-resistant CAL27 cells transduced with either scrambled shRNA or TPI1-targeting shRNA (shTPI1) were seeded into 96-well plates at a density of 5 × 10^3^ cells per well and allowed to adhere overnight. The next day, cells were pre-treated with the respective inhibitors for 2 h, followed by incubation for 24 h. DMSO was used as a vehicle control at equivalent concentrations.

### 2.6. Cell Viability Assay

Cell viability was assessed using the CellTiter-Glo luminescent cell viability assay (Promega, G7570, Fitchburg, WI, USA), according to the manufacturer’s instructions. Luminescence was measured with a SpectraMax iD5 plate reader (Molecular Devices, San Jose, CA, USA), and viability was normalized to the Scramble control group.

### 2.7. IC50 Assay

To assess the sensitivity of oral squamous cell carcinoma (OSCC) cells to cisplatin and erastin, cell viability assays were performed using the Cell Counting Kit-8 (CCK-8, Dojindo, Tokyo, Japan), and half-maximal inhibitory concentration (IC_50_) values were calculated. Parental and cisplatin-resistant CAL 27 cells, including TPI1 knockdown and overexpression groups, were seeded into 96-well plates at a density of 5 × 10^3^ cells per well and incubated overnight at 37 °C.

For cisplatin treatment, cells were exposed to increasing concentrations of cisplatin (0–8 μM) for 48 h. For erastin treatment, cells were exposed to varying concentrations of erastin (0–8 μM) for 24 h. After drug treatment, 10 µL of CCK-8 solution was added to each well, followed by a 2 h incubation at 37 °C. Absorbance was measured at 450 nm using a microplate reader (BioTek, Winooski, VT, USA). Cell viability was expressed as a percentage relative to untreated control cells. IC_50_ values were calculated by fitting the dose–response curves using nonlinear regression analysis in GraphPad Prism 9.0. All experiments were conducted in triplicate and independently repeated at least three times.

### 2.8. Creation of Cisplatin-Resistant Cell Lines and Transfection

The cisplatin-resistant cell line (CAR) was created through the clonal selection of CAL27, treated with 10–100 μM cisplatin for 10 cycles, followed by a recovery phase, based on Chang et al.’s research.

### 2.9. Transfection

shRNA targeting TPI1 (sh-TPI1) and a non-targeting control (Scramble) were sourced from GeneChem (Shanghai, China) (Table 1). Both were transfected into oral cancer cells using Lipofectamine 3000 (Invitrogen, Carlsbad, CA, USA) and incubated for 48 h. Transfection efficiency was assessed by qRT-PCR, and the transfected cells were then collected for further experiments.

### 2.10. Western Blot Assay

Protein was extracted using a cell lysis buffer (Cell Signaling Technology, Danvers, MA, USA) and resolved by 10% SDS-PAGE before being transferred to a PVDF membrane. The membranes were blocked with 5% skim milk for 1 h at 25 °C, then incubated overnight at 4 °C with the following primary antibodies: TPI1 (1:1000, 10713-1-AP), SLC7A11 (1:1000, 26864-1-AP), GPX4 (1:1000, 82822-2-RR), E-Cad (1:2000, 83991-4-RR), N-Cad (1:1000, 82968-1-RR), Snail (1:1000, 21350-1-AP), and GAPDH (10494-1-AP, all from Proteintech). The membranes were incubated for 1 h at 25 °C with an HRP-conjugated secondary antibody. Blots were detected using enhanced chemiluminescence and Western blotting reagents, with GAPDH serving as a loading control.

### 2.11. Reverse Transcription-Quantitative PCR (RT-qPCR)

Total RNA was extracted with an RNA extraction kit (Sigma-Aldrich) and reverse transcribed into cDNA using a reverse transcriptase kit (Sigma-Aldrich). RT-PCR was conducted with the SYBR^®^ Green Quantitative RT-PCR kit (Sigma-Aldrich) following the manufacturer’s instructions, using the Master cycler ep real plex detection system (Eppendorf, Hamburg, Germany). Primer sequences are listed in Table 2.

### 2.12. 3-[4,5-Dimethyl-2-thiazolyl]-2,5 diphenyl-2H-tetrazolium Bromide (MTT) Assay

Cell viability was assessed with the MTT kit (Sigma-Aldrich). Cells were cultured in 96-well plates for 24 h, followed by the addition of 20 μL MTT (2.5 mg/mL) and incubation at 37 °C for 4–6 h. Formazan crystals were dissolved in DMSO after removing the medium, and absorbance was measured at 460 nm using a SpectraMax M2 microplate reader (BioTek, Winooski, VT, USA).

### 2.13. Transwell Migration Assay

The migration and invasion of oral cancer cells were assessed using a transwell assay. Cells (3 × 10^4^) were placed in the upper chamber with serum-free RPMI1640, while the lower chamber contained a medium with 10% FBS. After 24 h at 37 °C, cells in the lower chamber were fixed with 4% paraformaldehyde and stained with 0.1% crystal violet. Then migrated cells were counted using a light microscope. 

### 2.14. Wound Healing Assays

For wound healing assays, wounds were created on a cell monolayer in a 6-well plate using 10 μL pipette tips until 95% of the cells were covered. Images were captured at 24 and 48 h, and the migration distance between the dotted lines was measured and normalized to control cells.

### 2.15. Fe^2+^ Measurement

Ultraviolet-visible (UV-vis) spectroscopy, utilizing three prevalent chromogenic reagents—namely ferrozine, 2,2′-bipyridine, and 1,10-phenanthroline—was employed for the detection and quantification of ferrous ions (Fe^2+^) and ferric ions (Fe^3+^). Following the knockdown of TPI1, cells were harvested, and the concentration of Fe^2+^ was assessed using the Solarbio BC5415 assay.

### 2.16. Measurement of ROS Production, GSH Level, and Lipid Peroxidation

Oral cancer cells were treated with or without TPI1 knockdown for 8 h. Cellular ROS and lipid ROS were measured using a ROS detection kit (Sigma-Aldrich), while glutathione (GSH) levels were assessed with a GSH colorimetric kit. Lipid peroxidation was evaluated by measuring the malondialdehyde (MDA) concentration using a lipid peroxidation detection kit (Sigma-Aldrich).

### 2.17. In Vivo Assay

Five-week-old athymic M-NSG mice were sourced from Beijing Vital River Laboratory. CAL27 cells (40 μL, 0.5 × 10^6^ cells) with sh-TPI1 and Scramble were injected subcutaneously into the bilateral flanks of mice (*n* = 4/group). To investigate TPI1’s role in cisplatin resistance in oral cancer cells, mice were assigned to Scramble, cisplatin, and shTPI1 + cisplatin groups (*n* = 4/group). Once tumor nodules appeared, mice in the cisplatin and shTPI1 + cisplatin groups received daily intraperitoneal injections of cisplatin (25 mg/kg). They were euthanized after 28 days. The tumor volume and weight were measured every 7 days. The tumor volume was calculated as (length × width^2^)/2. The animal research procedures were carried out in accordance with the China Animal Welfare Legislation and were approved by the Biomedical Ethics Committee of Peking University <Approval no: PUIRB-LA2024279>. This study is reported in accordance with ARRIVE guidelines. 

### 2.18. Statistical Analysis

We conducted statistical analyses with SPSS 22.0 and GraphPad Prism 8.0, using data from at least three independent experiments. Results are presented as mean ± standard deviation. Differences between groups were assessed using the Student’s *t*-test and chi-square test. Survival analysis was performed with the KM plotter method and log-rank test. Correlation was quantified using Spearman’s rank correlation coefficient (r). A *p*-value of <0.05 (two-tailed) was deemed statistically significant.

## 3. Results

### 3.1. High TPI1 Levels in Oral Cancer Are Associated with Poor Prognosis and Serve as an Independent Predictive Biomarker

To clarify TPI1’s role in oral cancer, we analyzed its expression and correlation with patient outcomes using data from TCGA and GEO. Our findings showed that TPI1 expression was significantly higher in cancer tissues than in normal oral tissues (Figure 1A). Additionally, TPI1 levels positively correlated with histologic grade, invasion depth, and lymph node metastasis (Figure 1B), and higher TPI1 expression was associated with worse overall survival (OS) in the cohort (Figure 1C). Cox regression analyses revealed that TPI1 expression is an independent prognostic factor for oral cancer outcomes (Figure 1D). Western blot and RT-qPCR analyses confirmed TPI1 expression changes in six paired oral cancer tissues (Figure 1E and Figure 1F, respectively), indicating that TPI1 may serve as a marker for oral cancer progression.

To understand TPI1’s role in oral cancer, we studied 107 patients and 28 normal tissues. Our analysis showed increased TPI1 in tumors, especially in invasive cases (Figure 1G), with lymph node involvement (Figure 1H) and metastasis (Figure 1I). High TPI1 levels were linked to worse overall and progression-free survival (Figure 1J). Taken together, these results suggest TPI1 may function as an oncogene in oral cancer.

### 3.2. TPI1 Knockdown Impairs Oral Cancer Prognosis Both In Vitro and In Vivo

To assess TPI1 function in oral cancer, we tested its levels in different cells and results showed that oral cancer expressed higher levels of TPI1 than HOK cells. (Figure 2A). We then studied TPI1’s role in oral cancer by silencing it in SCC15 cells, confirmed through Western blotting and RT-qPCR (Figure 2B,C). Given TPI1’s association with fibrosis and angiogenesis in hepatocarcinoma, we examined its impact on extracellular matrix processing. TPI1 knockdown decreased N-cadherin and Snail levels while increasing E-cadherin, indicating its role in promoting the epithelial–mesenchymal transition (EMT) (Figure 2C). Suppressing TPI1 decreased the proliferation, migration, and invasion of BC cells (Figure 2D–F). To investigate TPI1’s role in oral cancer progression in vivo, CAL27 cells with TPI1 knockdown were used (Figure 2G). A subcutaneous xenograft model in M-NSG mice showed that shTPI1 significantly reduced tumor weight compared with the Scramble groups, consistent with in vitro findings (Figure 2H,I). IHC analysis of Ki67 confirmed the knockdown efficiency (Figure 2J). These data suggest that TPI1 promotes oral cancer progression.

### 3.3. TPI1 Overexpression Accelerates Tumor Growth and Causes Cisplatin Resistance

We overexpressed TPI1 in SCC9 cells to demonstrate its role. Western blot and RT-qPCR confirmed TPI1 overexpression in oral cancer (Figure 3A,B). This upregulation significantly enhanced the proliferation and migration of ccRCC cells (Figure 3C,D), indicating that TPI1 promotes oral cancer progression. Cisplatin exposure upregulated TPI1 expression in CAL27 cells (Figure 3E,F), and cisplatin-resistant cells displayed elevated TPI1 levels even under moderate drug stimulation (Figure 3G,H). An IC50 assay indicated that TPI1 knockdown enhanced cisplatin efficacy (Figure 3I). In an in vivo tumor model, the cisplatin injection demonstrated that TPI1 inhibition improved treatment outcomes, reducing tumor weight and volume (Figure 3K–M). Ki67 IHC staining revealed that TPI1 knockdown combined with cisplatin most effectively inhibited tumor growth (Figure 3N). Taken together, we identified TPI1’s role in promoting oral cancer and aiding cisplatin resistance.

### 3.4. TPI1 Inhibition Triggered Ferroptosis in Cisplatin-Resistant Oral Cancer

To investigate the mechanism of TPI1-induced cisplatin resistance in oral cancer, we performed RNA-Seq following TPI1 silencing in cisplatin-resistant cells (Figure 4A). The results indicated that ferroptosis was the most significantly affected pathway (Figure 4B), highlighting its role in regulating cell death in oral cancer. Additionally, we observed changes in classical GPX4-related pathway proteins post-TPI1 knockdown (Figure 4C), prompting further research. Western blot and RT-qPCR analyses revealed decreased levels of SLC7A11 and GPX4, while TFRC was upregulated after TPI1 knockdown (Figure 4D,E). Our investigation revealed that silencing TPI1 in cisplatin-resistant CAL27 cells resulted in increased levels of Fe^2+^ (Figure 4F), along with elevated lipid-ROS (Figure 4G) and lipid peroxidation (Figure 4H), thereby corroborating our previous findings. Additionally, GSH levels were reduced in these cells (Figure 4I).

To further confirm the role of ferroptosis in mediating the cytotoxic effect of TPI1 knockdown, we conducted phenotypic, biochemical, and pharmacological rescue assays in cisplatin-resistant CAL27 cells. Morphological observation showed that TPI1 knockdown induced substantial cell shrinkage and loss of adhesion, which were visibly reversed upon treatment with the ferroptosis inhibitor ferrostatin-1 (Fer-1) (Figure 4J). Quantification of lipid ROS levels revealed that Fer-1 significantly reduced lipid peroxidation in TPI1-silenced cells (Figure 4K), supporting the involvement of ferroptosis. Moreover, TPI1 knockdown increased erastin sensitivity in a dose-dependent manner, as demonstrated by a reduced IC_50_ value compared with the control cells (Figure 4L). To validate the specific cell death pathway involved, we compared the effects of cell death inhibitors. Among ferroptosis (Fer-1), apoptosis (Z-VAD-FMK), necroptosis (Nec-1), and autophagy (3-MA) inhibitors, only Fer-1 significantly rescued the viability of TPI1-depleted cells, whereas other inhibitors showed no notable protective effect (Figure 4M). These findings suggest that TPI1 knockdown promotes ferroptosis in cisplatin-resistant cells, representing a potential therapeutic vulnerability.

### 3.5. TPI1 Overexpression Protects Against Erastin-Induced Ferroptosis in Oral Cancer Cells

To further validate the role of TPI1 as a suppressor of ferroptosis, we examined whether TPI1 overexpression could counteract ferroptosis induction. Cisplatin-resistant CAL27 cells were transfected with a TPI1-overexpressing construct and subsequently treated with 5 μM erastin. Cell viability assays demonstrated that TPI1-overexpressing cells exhibited significantly higher survival rates compared with vector control cells under erastin treatment (Figure 5A), indicating enhanced resistance to ferroptosis. Further biochemical assays confirmed that TPI1 overexpression significantly reduced intracellular ferrous iron (Fe^2+^) levels (Figure 5B) and markedly decreased lipid reactive oxygen species (lipid-ROS) accumulation following erastin exposure (Figure 5C). These data suggest that TPI1 overexpression inhibits ferroptosis by restricting iron availability and suppressing lipid peroxidation.

Taken together, these findings support the conclusion that elevated TPI1 expression enables cisplatin-resistant oral cancer cells to escape ferroptotic cell death. By maintaining redox balance and limiting ferroptosis-associated stress, TPI1 contributes to therapy resistance. Therefore, targeting TPI1 may represent a promising therapeutic strategy to restore ferroptotic vulnerability and overcome chemoresistance in oral cancer (Figure 6).

## 4. Discussion

Despite advancements in diagnostic strategies and therapeutic modalities, oral squamous cell carcinoma (OSCC) continues to exhibit poor clinical outcomes, largely due to the development of chemoresistance [3,28,29]. This persistent obstacle underscores the need to identify new therapeutic vulnerabilities in drug-resistant tumors. In this study, we uncover a previously unrecognized role for TPI1, a glycolytic enzyme, in regulating ferroptosis—a form of iron-dependent non-apoptotic cell death—and demonstrate its significance in mediating resistance to cisplatin in OSCC.

TPI1 has been historically characterized as a metabolic enzyme involved in glycolysis, catalyzing the reversible interconversion of glyceraldehyde-3-phosphate and dihydroxyacetone phosphate [22,25]. However, recent studies have revealed its involvement in cancer progression beyond metabolic regulation, including roles in cell proliferation, metastasis, and redox homeostasis [21]. Our findings expand on these observations by illustrating that TPI1 also plays a suppressive role in ferroptosis, a death pathway gaining increasing attention for its potential to eliminate therapy-resistant tumor cells.

Mechanistically, we show that TPI1 is significantly upregulated in cisplatin-resistant OSCC cells and tissues. Its silencing not only inhibits tumor cell proliferation, migration, and invasion, but also activates ferroptosis, as evidenced by the increased accumulation of intracellular Fe^2+^, elevated lipid-ROS, increased MDA levels, and depletion of GSH. These cellular events indicate a breakdown in the antioxidant defense system, specifically through the downregulation of two key ferroptosis-suppressing molecules: GPX4 and SLC7A11. This disruption makes the cancer cells more susceptible to lipid peroxidation and subsequent ferroptotic death.

Importantly, we found that the ferroptosis inhibitor ferrostatin-1, but not inhibitors of other forms of cell death, effectively rescued the cell death phenotype induced by TPI1 knockdown. This specificity strongly supports the conclusion that TPI1 inhibition sensitizes resistant OSCC cells to ferroptosis rather than to other forms of cell death such as apoptosis or necroptosis. Additionally, TPI1-silenced cells demonstrated increased sensitivity to erastin, a well-established ferroptosis inducer. Together, these findings position TPI1 as a key negative regulator of ferroptosis in cisplatin-resistant OSCC. We also found that overexpression of TPI1 could resist the occurrence of ferroptosis.

From a therapeutic standpoint, our data suggest that targeting TPI1 could serve as a dual-function strategy—both reversing drug resistance and promoting ferroptosis-mediated cell death. Given the association between TPI1 expression and EMT markers, TPI1 may also facilitate metastatic potential in addition to therapy resistance. Therefore, inhibiting TPI1 may not only sensitize tumors to cisplatin but also mitigate their invasive properties, making it a particularly attractive target for combination therapies involving ferroptosis inducers and conventional chemotherapeutic agents.

## 5. Conclusions

In conclusion, the present study suggests that TPI1 may be a novel regulator of ferroptosis and is also a potential therapeutic target in cisplatin-resistant oral cancer.

### Limitations

While our findings offer novel insights into the role of TPI1 in ferroptosis and chemoresistance, some aspects require further exploration. The precise molecular mechanisms by which TPI1 regulates ferroptosis-related genes remain to be clarified, and additional validation in broader clinical cohorts and pharmacological models will be beneficial to strengthen the translational relevance of this work.

## Figures and Tables

**Figure 1 biomedicines-13-01225-f001:**
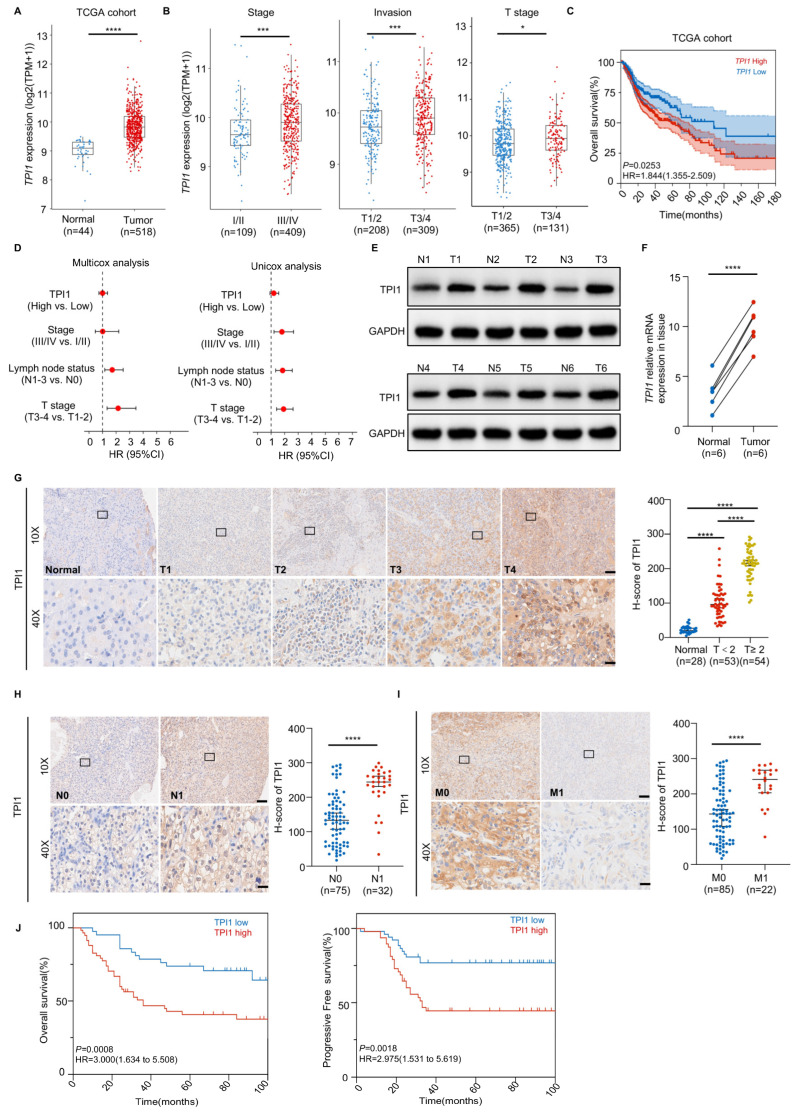
High TPI1 expression is associated with poor prognosis in oral cancer patients. (**A**) TPI1 mRNA expression levels in oral cancer vs. adjacent non-tumor tissues were analyzed using TCGA datasets. (**B**) TPI1 expression stratified by clinical stage, invasion depth, and histological grade in TCGA cohort. (**C**) Kaplan–Meier survival curves for overall survival (OS) based on high vs. low TPI1 expression (TCGA cohort). (**D**) Univariate and multivariate Cox regression analyses assessing TPI1 and clinical features as predictors of OS. (**E**,**F**) Western blot (**E**) and RT-qPCR (**F**) analyses of TPI1 in six paired clinical oral tumors and adjacent tissues. (**G**–**I**) Immunohistochemical (IHC) analysis of TPI1 in 107 oral cancer samples across T (**G**), N (**H**), and M (**I**) stages. (**J**) Kaplan–Meier plots comparing OS and progression-free survival (PFS) between high vs. low TPI1 expression groups. Scale bars: 500 μm (overview images) and 50 μm (magnified fields). Data are presented as mean ± standard deviation (SD). Statistical comparisons between two groups were performed using an unpaired two-tailed Student’s *t*-test. For survival analyses, the log-rank (Mantel–Cox) test was used; *p* < 0.05 (*), *p* < 0.01 (**), *p* < 0.001 (***), *p* < 0.0001 (****). The number of biological replicates (*n*) for each experiment is indicated in the corresponding figure’s panels or legends.

**Figure 2 biomedicines-13-01225-f002:**
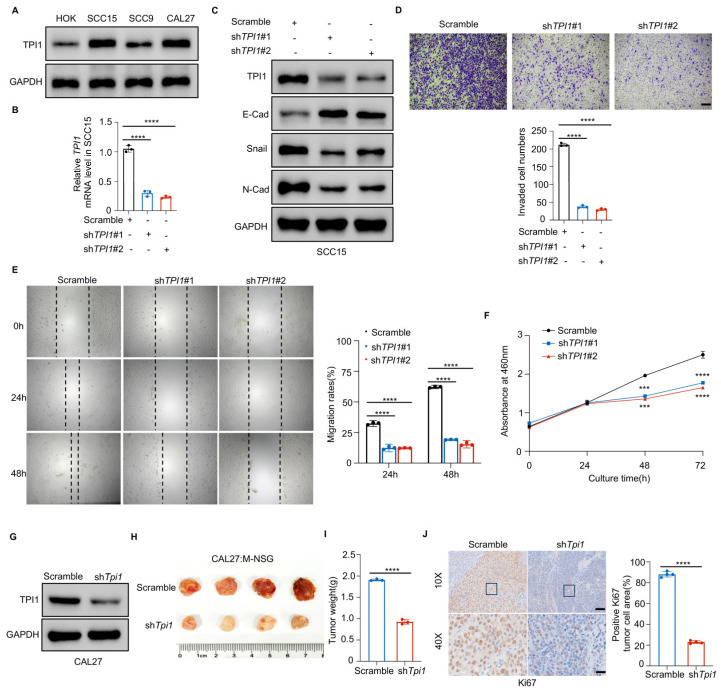
TPI1 knockdown reduces oral cancer growth and metastasis. (**A**) TPI1 expression levels across oral epithelial (HOK) and cancer cell lines (SCC15, SCC9, CAL27). (**B**,**C**) Validation of TPI1 knockdown in SCC15 cells via RT-qPCR (**B**) and Western blot (**C**). EMT markers were also evaluated. (**D**–**F**) Quantification of migration (**D**), invasion (**E**), and proliferation (**F**) in SCC15 cells with scrambled or shTPI1 vectors. (**G**–**I**) TPI1 knockdown was confirmed in CAL27 cells (**G**), which were used for in vivo xenograft experiments in M-NSG mice (*n* = 4/group). Tumor weights were measured on Day 28 (**I**). (**J**) Ki67 IHC staining of xenografts to assess proliferative activity. Scale bars: 500 μm (low magnification) and 50 μm (high magnification). Data are presented as mean ± standard deviation (SD). Statistical comparisons between two groups were performed using an unpaired, two-tailed Student’s *t*-test. *p* < 0.05 (*), *p* < 0.01 (**), *p* < 0.001 (***), *p* < 0.0001 (****). The number of biological replicates (*n*) for each experiment is indicated in the corresponding figure panels or legends.

**Figure 3 biomedicines-13-01225-f003:**
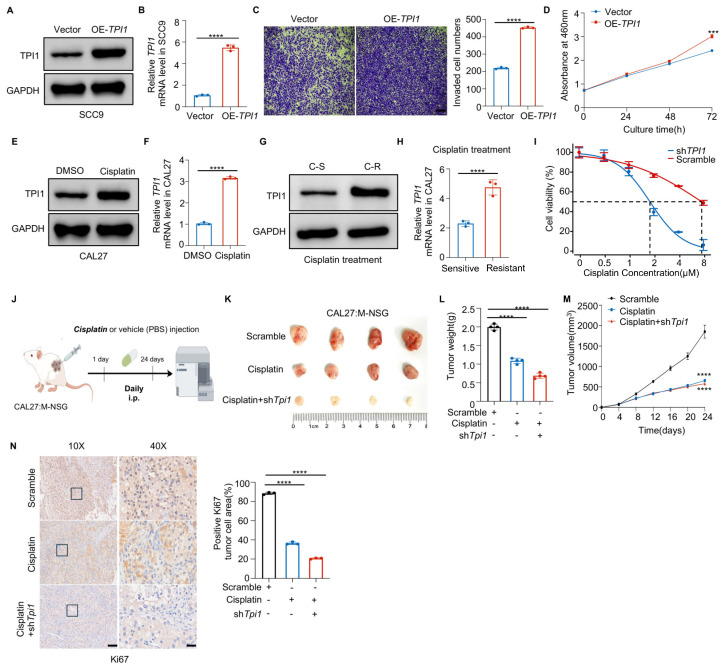
Increased TPI1 levels drive tumor growth and boost cisplatin resistance. (**A**,**B**) Validation of TPI1 overexpression in SCC9 cells via Western blot (**A**) and RT-qPCR (**B**). (**C**,**D**) Quantitative analysis of cell migration (**C**) and proliferation (**D**) following TPI1 overexpression. (**E**,**F**) Assessment of TPI1 levels in CAL27 cells with or without cisplatin treatment (10 μM) by Western blot (**E**) and RT-qPCR (**F**). (**G**,**H**) Comparison of TPI1 expression between cisplatin-sensitive (C-S) and -resistant (C-R) CAL27 cells under drug exposure. (**I**) IC_50_ assay measuring cisplatin sensitivity in CAL27 cells ± shTPI1. (**J**–**M**) In vivo tumor volume (**L**) and weight (**M**) measurements in xenograft mice (*n* = 4/group) following treatment. (**N**) Ki67 IHC staining of tumors from the indicated treatment groups. Scale bars: 500 μm (left) and 50 μm (right). Data are presented as mean ± standard deviation (SD). Statistical comparisons between two groups were performed using an unpaired two-tailed Student’s *t*-test. *p* < 0.05 (*), *p* < 0.01 (**), *p* < 0.001 (***), *p* < 0.0001 (****). The number of biological replicates (*n*) for each experiment is indicated in the corresponding figure’s panels or legends.

**Figure 4 biomedicines-13-01225-f004:**
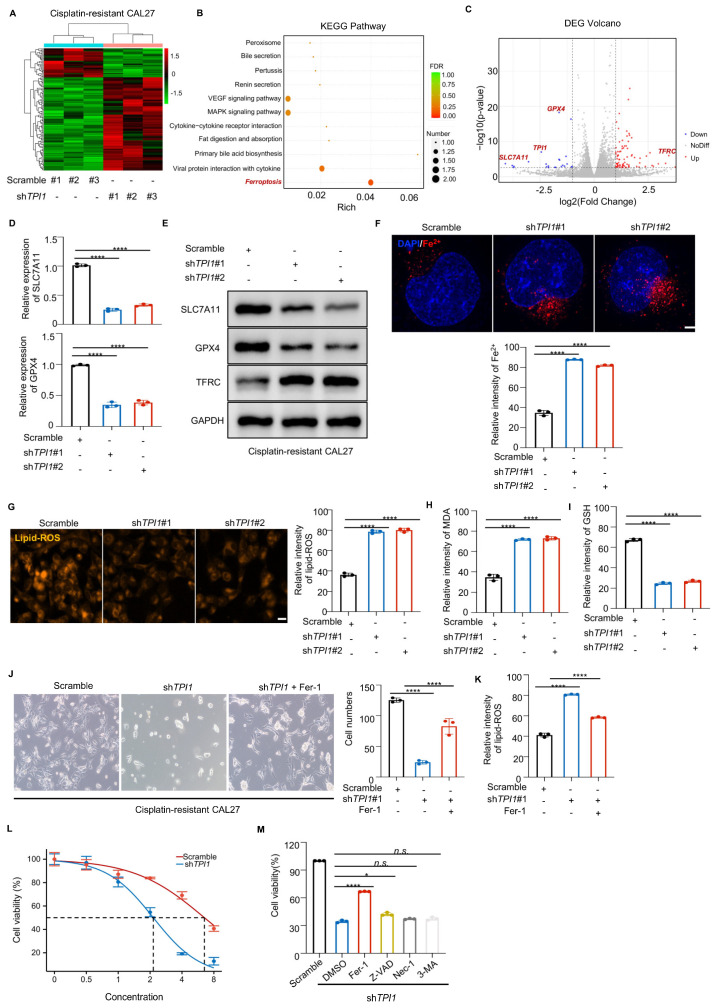
TPI1 silencing induces ferroptosis in oral cancer cells in vitro. (**A**) RNA-Seq of TPI1-knockdown CAL27 cells revealed ferroptosis as a significantly enriched pathway. (**B**) KEGG pathway enrichment analysis based on differentially expressed genes. (**C**) Volcano plot of gene expression changes upon TPI1 knockdown. (**D**,**E**) RT-qPCR (**D**) and Western blot (**E**) analysis of SLC7A11, GPX4, and TFRC following TPI1 silencing. (**F**–**I**) Biochemical assays of ferroptosis markers: intracellular Fe^2+^ (**F**), lipid ROS (Scale bar: 400 nm) (**G**), malondialdehyde (MDA) (Scale bar: 20 μm) (**H**), and glutathione (GSH) levels (**I**) in CAL27 cells. (**J**) Bright-field images showing cell morphology in the Scramble control, shTPI1, and shTPI1 + Fer-1-treated CAL27 cells. (**K**) Quantification of lipid-ROS in cells treated as in (**J**), with Fer-1 significantly reducing lipid peroxidation in TPI1-depleted cells. (**L**) IC_50_ dose-response curves comparing erastin sensitivity in Scramble and shTPI1-transfected CAL27 cells. TPI1 knockdown notably enhances erastin sensitivity. (**M**) Rescue experiment assessing the effects of cell death inhibitors on TPI1-knockdown cells. Among ferroptosis (Fer-1, 2 μM), apoptosis (Z-VAD-FMK, 20 μM), necroptosis (Nec-1, 20 μM), and autophagy (3-MA, 5 mM) inhibitors, only Fer-1 significantly restored cell viability. Values represent mean ± SD. Data are presented as mean ± standard deviation (SD). Statistical comparisons between two groups were performed using an unpaired two-tailed Student’s *t*-test. *p* < 0.05 (*), *p* < 0.01 (**), *p* < 0.001 (***), *p* < 0.0001 (****). The number of biological replicates (*n*) for each experiment is indicated in the corresponding figure’s panels or legends.

**Figure 5 biomedicines-13-01225-f005:**
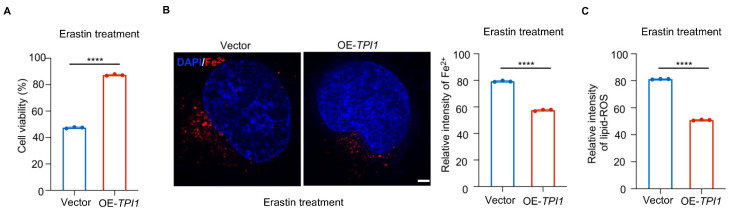
TPI1 blocks erastin-induced ferroptosis in oral cancer cells. (**A**) Cell viability of cisplatin-resistant CAL27 cells transfected with vector control or TPI1-overexpressing plasmid after treatment with 5 μM erastin for 24 h. (**B**) Quantification of intracellular Fe^2+^ levels following erastin treatment in control and TPI1-overexpressing cells. (Scale bar: 400 nm) (**C**) Lipid-ROS levels were measured using C11-BODIPY staining. Statistical comparisons between two groups were performed using an unpaired two-tailed Student’s *t*-test. *p* < 0.05 (*), *p* < 0.01 (**), *p* < 0.001 (***), *p* < 0.0001 (****). The number of biological replicates (*n*) for each experiment is indicated in the corresponding figure’s panels or legends.

**Figure 6 biomedicines-13-01225-f006:**
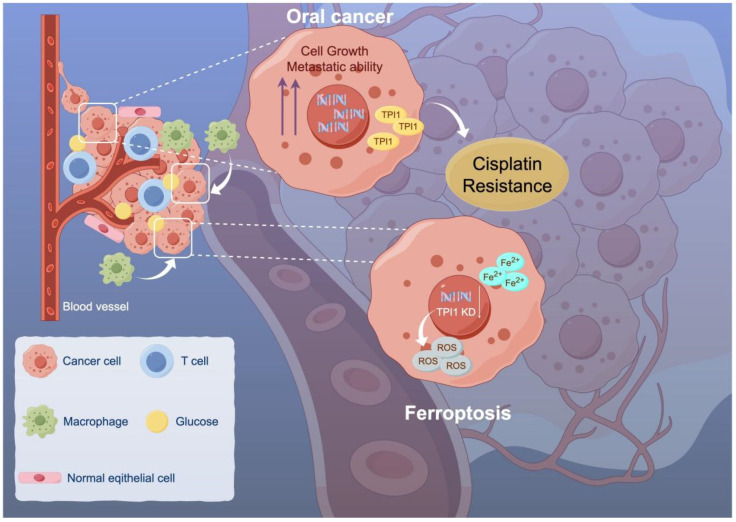
Inhibition of TPI1 promotes cisplatin sensitivity. Oral cancer’s growth and metastasis are driven by high TPI1 expression, which also induces cisplatin resistance. TPI1 further regulates EMT progression. Inhibition of TPI1 is a potential treatment target for cisplatin-resistant patients.

**Table 1 biomedicines-13-01225-t001:** Target Sequence.

	Target Sequence
sh-TPI1	F:CCGGTGATGTGGATGGCTTCCTTGTCTCGAGACAAGGAAGCCATCCACATCATTTTTG
	R:AATTCAAAAATGATGTGGATGGCTTCCTTGTCTCGAGACAAGGAAGCCATCCACATCA
Scramble	F:CCGGTTGGTGCTATGCGTGTACTGTCTCGAGAGTACACGCATAGCACCAATTTTTG
	R:AATTCAAAAATTGGTGCTATGCGTGTACTGTCTCGAGAGTACACGCATAGCACCAA

**Table 2 biomedicines-13-01225-t002:** Target Sequence.

	Target Sequence
TPI1	F: ACTGCCTATATCGACTTCGCC
	R: AAGCCCCATTAGTCACTTTGTAG

## Data Availability

The data used to support the findings of this study are available from the corresponding author upon request.

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
