# Peer review of "Inhibition of TPI1 Sensitizes Cisplatin-Resistant Oral Cancer to Ferroptosis"

_biomedicines, 2025, doi:10.3390/biomedicines13051225_

Round 1
Reviewer 1 Report
Comments and Suggestions for Authors
The authors (Dandan Wang and colleagues) demonstrated that inhibition of triosephosphate isomerase 1 (TPI1) sensitizes cisplatin-resistant oral cancer cells to ferroptosis. Their key conclusion is that this outcome is mediated by suppressed cell proliferation, migration, and invasion, driven by ferroptosis through increased reactive oxygen species levels and lipid peroxidation in cisplatin-resistant cell lines. Although the manuscript is generally well-organized and presents novel data with a solid methodology, the text and figures should be thoroughly revised to improve clarity and readability. I believe these findings will offer valuable insights for researchers in this field. The authors should consider the following issues to further strengthen the manuscript:
Comments:
- This is a very interesting finding; however, the authors need to conduct additional experiments to confirm that ferroptosis is the sole contributor to the increased cisplatin sensitivity observed upon TPI1 inhibition, as this appears to be the only novel finding presented in the manuscript. Specifically, they should examine the effects of well-established ferroptosis inhibitors commonly used in cancer research. It is essential to demonstrate that blocking ferroptosis diminishes the response of TPI1-silenced CAL27 cells, both with and without cisplatin treatment. Additionally, the authors should assess EMT markers and evaluate cell proliferation, migration, and invasion under these conditions to gain a more comprehensive understanding of the underlying mechanisms.
- Authors should provide the full form of any abbreviation the first time it is used in the manuscript, such as Epithelial-Mesenchymal Transition (EMT) at line 30.
- The sentence “Additionally, AEBP1 silencing may hinder the EMT pathway” (line 30) appears both redundant and unclear, as neither AEBP1 nor the EMT pathway is introduced or explained earlier in the abstract. It is recommended that the authors provide a brief background for both terms before referencing their potential roles. More importantly, the manuscript does not present any original research or data related to AEBP1, making this statement unsupported and irrelevant in the current context. It is advised that the authors remove or revise this sentence accordingly.
- I would like to request a revision of the sentence at lines 190 - 191: "the reader to read the text with align on the figure RT-qPCR and Western blot analyses (Fig. 1E–F)", as the alignment of figure references is incorrect. The RT-qPCR results are shown in Fig. 1F, whereas the Western blot results are in Fig. 1E. Please update the text to reflect the correct figure alignment.
- There are several formatting mistakes, such as 'Investigated' (not 'investi-gated') at line 20, and 'proliferation' (not 'pro-liferation'). I suggest the authors carefully review the manuscript for such errors throughout.
Reviewer 2 Report
Comments and Suggestions for Authors
The authors explore the role of TPI1 in cisplatin-resistant oral cancer by focusing on ferroptosis as a therapeutic mechanism. The manuscript is presented well but requires major revision before consideration for publication. The following points need to be addressed:
- Briefly explain how the author developed the cisplatin-resistant cell line and the cutoff selected to designate CAL27 cells as cisplatin-resistant.
- Experiments with ferroptosis inhibitors further strengthen the study; however, cite from the literature the impact of ferroptosis inhibitors on selected markers like Fe2+, ROS, and MDA levels.
- Molecular mechanism studies need to be in-depth. The use of RNAseq or Proteomics data might be helpful to connect TPI1 with ferroptosis. Need to add or address using literature.
- Figure legend should be written in a proper way with p-value, error bars, etc.
- Discussion needs to be rewritten properly, as the majority of the results are written again in the discussion part.
- Grammar or English clarity needs to be addressed in the overall manuscript to clarify the text.
English needs to be improved for clarity.
Round 2
Reviewer 1 Report
Comments and Suggestions for Authors
The authors have adequately addressed the concerns in their revised manuscript, and it may now be suitable for acceptance and publication. The only issue I have found is that some words or sentences appear as hyperlinks, for example, between lines 47 and 55. I would like to ask the authors to carefully revise the manuscript to address and correct these issues throughout.
Author Response
Dear Editor,
Thank you very much for your careful review and constructive comments.
We have carefully checked the entire manuscript and removed all unintended hyperlinks, including those appearing between lines 47 and 55 as you kindly pointed out. We have ensured that there are no remaining hyperlinks throughout the revised manuscript.
We appreciate your attention to detail and the opportunity to further improve our submission.
Thank you again for your consideration.
Reviewer 2 Report
Comments and Suggestions for Authors
The author provided point-to-point revision and significantly improved the manuscript, However, a few concerns need to be addressed:
- References are missing in the overall manuscript due to unlinked endnote files.
- Still need to support the data correlating that TPI suppresses Ferroptosis mechanistically.
- Kindly add the IC50 value of parental CAL 27 cells and the resistant cells.
- Still introduction and discussion need English language editing.
- Figure needs statistical description improvement.
- What are the limitations of this study that need to be addressed in the conclusion?
Introduction and discussion need English language editing
